# Nonstoichiometric Cu_0.6_Ni_0.4_Co_2_O_4_ Nanowires as an Anode Material for High Performance Lithium Storage

**DOI:** 10.3390/nano10020191

**Published:** 2020-01-22

**Authors:** Junhao Li, Ningyi Jiang, Jinyun Liao, Yufa Feng, Quanbing Liu, Hao Li

**Affiliations:** 1School of Chemical Engineering and Light Industry, Guangdong University of Technology, Guangzhou 510006, China; ljh15626204622@163.com; 2School of Chemical Engineering and Technology, Tianjin University, Tianjin 300072, China; nyjiang18@163.com; 3Tianjin Institute of Power Sources, Tianjin 30084, China; 4Tianjin Space Power Technology Co., Ltd., Tianjin 30084, China; 5School of chemistry and Materials Engineering, Huizhou University, Huizhou 516007, China; jyliao@126.com (J.L.); yufafeng@126.com (Y.F.)

**Keywords:** lithium-ion battery, anode material, nonstoichiometric, doping, Cu_0.6_Ni_0.4_Co_2_O_4_ nanowires

## Abstract

Transition metal oxide is one of the most promising anode materials for lithium-ion batteries. Generally, the electrochemical property of transition metal oxides can be improved by optimizing their element components and controlling their nano-architecture. Herein, we designed nonstoichiometric Cu_0.6_Ni_0.4_Co_2_O_4_ nanowires for high performance lithium-ion storage. It is found that the specific capacity of Cu_0.6_Ni_0.4_Co_2_O_4_ nanowires remain 880 mAh g^−1^ after 50 cycles, exhibiting much better electrochemical performance than CuCo_2_O_4_ and NiCo_2_O_4_. After experiencing a large current charge and discharge state, the discharge capacity of Cu_0.6_Ni_0.4_Co_2_O_4_ nanowires recovers to 780 mAh g^−1^ at 50 mA g^−1^, which is ca. 88% of the initial capacity. The high electrochemical performance of Cu_0.6_Ni_0.4_Co_2_O_4_ nanowires is related to their better electronic conductivity and synergistic effect of metals. This work may provide a new strategy for the design of multicomponent transition metal oxides as anode materials for lithium-ion batteries.

## 1. Introduction

Lithium-ion batteries (LIB) have many advantages, such as high energy density, excellent cycle stability, no memory effect, and so on [1]. They are widely used in many fields, such as electronic products, electric vehicles, and energy storage system, which have greatly improved modern humans’ lives [2]. The energy density of LIB is closely related to the lithium storage performance and voltage of the anode materials. The transition metal oxide Co_3_O_4_ used as a promising anode material for lithium-ion batteries, first reported by Poizot [3] in 2000, has attracted extensive interest of researchers with 890 mAh g^−1^ theoretical capacity, which is much higher than commercial graphite (372 mAh g^−1^). Different from lithium intercalation mechanism of graphite, Co_3_O_4_ is based on a conversion reaction as anode materials transforms into metallic cobalt, with great volume expansion and metal reunion during lithiation/deithiation [4].

Due to the inherent defects of the material, many researchers attempt to enhance the electrochemical performance of Co_3_O_4_ by controlling their microstructures and optimizing their components [5]. On the one hand, great efforts have been devoted to the morphological control of Co_3_O_4_, and therefore, many of the different shaped Co_3_O_4_ nanostructures, such as peapod-like [6], yolk-shell [7], hollow nanofibers [8], nanowires [9,10], other complex structures [11,12,13,14,15,16,17,18] have been developed and applied as anode materials in lithium-ion batteries. Compared with other nanostructures, nanowires have large open space, which allows electrolytes to diffuse into the electrode inner region and shorten the solid-phase diffusion path of lithium ions. Moreover, the mechanical flexibility and Young’s modulus of nanowires are significant for micro-flexible electronic components [19]. For example, Guan [9] synthesized 1D Co_3_O_4_ mesoporous nanowires, grown on Ni foam, whose specific capacity was approximately 1600 mAh g^−1^ at current density of 0.5 A g^−1^. Zeng [20] designed quasi-single-crystalline mesoporous Co_3_O_4_ and CoO nanowires from precursor Co(Co_3_)_0.5_(OH) and the CoO nanowires remain nearly 100% capacity retention after 70 cycles.

On the other hand, it is a viable strategy to design transition-metal oxide composites for enhancing lithium storage performance [21]. It has been reported that the composites of bimetal oxide showed much higher electrochemical performance than the single ones due to low activation energy for electron transfer between cations [10,22,23]. MCo_2_O_4_ (M = Ni [24,25,26,27], Cu [28], Fe [29], Zn [30], Mn [31]) have been widely used as anode materials over the last few years. Lou [26] synthesized NiCo_2_O_4_ complex hollow spheres as anode material in lithium-ion battery and its specific capacity was 1400 mAh g^−1^ at 150 mA g^−1^ with great cycling stability. Jiang [32] reported CuCo_2_O_4_ nanoparticle showed high reversible capacity of 1040 mAh g^−1^ at 0.1 C. Cao [33] designed ultrathin ZnCo_2_O_4_ nanosheets for lithium anode with excellent long life (about 900 mAh g^−1^ after 200 cycles at 200 mA g^−1^ current density).

Herein, we report nonstoichiometric Cu_0.6_Ni_0.4_Co_2_O_4_ nanowires, which can be regarded as a composite of CuCo_2_O_4_ and NiCo_2_O_4_, as an anode material in lithium-ion battery. As we know, such composites, as an anode material in lithium-ion area has not been reported yet. Owing to the relatively low activation energy for electron transfer between Cu^2+^/Cu, Ni^2+^/Ni^3+^ and Co^2+^/Co^3+^, the electronic conductivity and cycling stability for Cu_0.6_Ni_0.4_Co_2_O_4_ nanowires are significantly improved. In the first cycle, discharge and charge capacities are 1120 mAh g^−1^ and 972 mAh g^−1^ at 50 mA g^−1^, respectively. After 50 cycles, the specific capacity of Cu_0.6_Ni_0.4_Co_2_O_4_ nanowires remain 880 mAh g^−1^ and the coulombic efficiency is approximately 100%. 

## 2. Experimental

### 2.1. Synthesis of Anode Materials

All the reagents (Tianjin Damao Chemical Reagent Factory, Tianjin, China) were analytically pure without further treatment. Similar preparation methods could be referred to our previous work [34]. To prepare the Cu_0.6_Ni_0.4_Co_2_O_4_ nanowires, 4 mmol cobalt acetate was dissolved in 30 mL ultrapure water. Subsequently, 4 mmol lauryl sodium sulfate, 1.2 mmol copper chloride, and the 0.8 mmol nickel chloride were added to the above cobalt acetate solution in sequence. Then, 48 mmol hexamethylenetetramine was dissolved in 30 mL ultrapure water to form a transparent solution which was dropped into the above mixed solution slowly. After stirring for 1 h, the resultant solution was poured into a 100 mL Teflon-lined stainless autoclave. Then the autoclave was heated at 120 °C for 12 h in oven. The obtained sediment was filtered, rinsed and dried. Finally, after calcination of the powder at 600 °C for 2 h in muffle, Cu_0.6_Ni_0.4_Co_2_O_4_ was obtained. Similarly, NiCo_2_O_4_ and CuCo_2_O_4_ were obtained by adjusting the addition amount of copper chloride and nickel chloride.

### 2.2. Characterizations

Crystal information of the samples was acquired by Bruker D8 Discover X-ray diffractometer (XRD, Tokyo, Japan) with Cu Kα radiation (λ = 1.5406Å). Microstructures, particle sizes and element mapping were analyzed by using a JEOL-7100F field-emission scanning electron microscope (FESEM, Hitachi, Japan). Interior structure was observed by a JEOL JEM-2100F transmission electron microscope (TEM, FEI, Hillsboro, OR, USA). The elements and chemical states of the sample were analyzed by a Kratos Axis Ultra DLD X-ray photoelectron spectrometer (XPS, VG, Manchester, UK). The molar ratios were determined with an Agilent 7800 Inductively Coupled Plasma (ICP) Mass Spectrometry (Thermo Fisher, Waltham, MA, US). 

### 2.3. Electrochemical Performance Test

The 2032-coin battery is assembled in a glove box filled with inert atmosphere for electrochemical performance test. The fabricated electrode consisted of active materials, acetylene black, polyvinylidene fluoride (PVDF) with a mass ratio of 7:2:1. Each component was well mixed to form a paste, which was coated on copper foil current collector. The loading on the electrode is approximately 1 mg cm^−2^. The electrolyte used was 1.0 M LiPF_6_ in a 50:50 (w/w) mixture of ethylene carbonate (EC) and diethyl carbonate (DEC). The specific capacity of the battery was tested by cyclic charge and discharge between 0–3 V (Land CT2001A, Hubei, China). The cyclic voltammetry and electrochemical impedance test were obtained by using an electrochemical workstation (CHI 760E, CH Instrument Ins., Shanghai, China). 

## 3. Results and Discussion

The X-ray diffraction (XRD) patterns of the NiCo_2_O_4_, CuCo_2_O_4_, and Cu_0.6_Ni_0.4_Co_2_O_4_ powders are shown in Figure 1. It is discovered that the characteristic peaks of the NiCo_2_O_4_ and CuCo_2_O_4_ are very similar. All the diffraction peaks of the NiCo_2_O_4_, CuCo_2_O_4_ powders correspond to the (220), (311), (222), (400), (422), (511), and (440) plane reflections of cubic spinel NiCo_2_O_4_ or CuCo_2_O_4_. Cu_0.6_Ni_0.4_Co_2_O_4_ has similar characteristic peaks as NiCo_2_O_4_ [35] and CuCo_2_O_4_. We can regard Cu_0.6_Ni_0.4_Co_2_O_4_ as a composite of NiCo_2_O_4_ and CuCo_2_O_4_ [36]. The peak at 25° for anode materials is attributed to the disorderedly stacked carbon, derived from incomplete decomposition of organic ingredients [37]. There is a small peak in NiCo_2_O_4_ at 44° indexing as NiO (JCPDS NO. 65-2901), which is also found in previous literature [38]. The peak at 33° for CuCo_2_O_4_ may be indexed as CuO (JCPDS NO. 44-0706). According to the Scherer’s formula, the average crystal size of a material can be calculated [39,40],
(1)D=0.89λ/(βcosθ)
where *D* is the average gain size, λ is X-ray wavelength and constant of 0.154 nm, β is half-height width of diffraction peak, and θ is Bragg diffraction angle. The average crystalline sizes of NiCo_2_O_4_ and CuCo_2_O_4_ are 16.5, and 24.9 nm, respectively. Meanwhile, that of Cu_0.6_Ni_0.4_Co_2_O_4_ is 18.0 nm, which is between 16.5 nm and 24.9 nm, indicating that the formation of composite of NiCo_2_O_4_ and CuCo_2_O_4_ will not result in a pronounced change in crystalline size.

Scanning electron microscopy (SEM) images are used to observe the morphology and structures of materials, as shown in Figure 2. Apparently, the samples of the NiCo_2_O_4_ and Cu_0.6_Ni_0.4_Co_2_O_4_ consist of nanowires with an average diameter of ca. 50 nm. Interestingly, all these nanowires are assembled by small nanoparticles. The structure of CuCo_2_O_4_ is nanoparticles with an average diameter of ca. 30 nm. The TEM images of three anode materials in Figure 3, present the inner structures for NiCo_2_O_4_, Cu_0.6_Ni_0.4_Co_2_O_4_, and CuCo_2_O_4_. The anode materials are solid instead of hollow. Besides, inner structure of Cu_0.6_Ni_0.4_Co_2_O_4_ is similar with NiCo_2_O_4_ rather than CuCo_2_O_4_. Figure 3g is high resolution transmission electron microscope (HRTEM) images of the Cu_0.6_Ni_0.4_Co_2_O_4_. The lattice spacing of ca. 0.23 nm, 0.2 nm, 0.28 nm, and 0.24 nm can be seen, corresponding to the crystal surfaces (222), (400), (220), and (311) planes of cubic spinel Cu_0.6_Ni_0.4_Co_2_O_4_. Element mapping of Cu_0.6_Ni_0.4_Co_2_O_4_ is measured in Figure 4. The elements of Co, Ni, Cu, and O can be found in the mapping picture, indicating that these elements are uniformly distributed. The result of ICP measure shows the ratio of Cu, Ni, and Co is 0.61:0.41:1.99, which are well match with the designed values.

In order to investigate the elements and chemical states on the surface of the anode materials, XPS analysis is carried out and the results are shown in Figure 5. In the XPS survey spectrum of Cu_0.6_Ni_0.4_Co_2_O_4_ (Figure 5c), the signals of elements of Co, Ni, and Cu can be observed, which was different from those of NiCo_2_O_4_ and CuCo_2_O_4_ (Figure 5a,b), as the former has both Cu and Ni elements, and the three transition elements Cu, Ni, and Co will affect the outer electronic structure of each other. As shown in Figure 5a,b, no signal of Ni is found in CuCo_2_O_4_, while the Cu signal was absent in NiCo_2_O_4_, which are as expected. Figure 5d–f is the Ni 2p, Cu 2p, and Co 2p spectrum of the anode materials, respectively. There are six peaks in Ni 2p spectrum for NiCo_2_O_4_ and Cu_0.6_Ni_0.4_Co_2_O_4_. The peaks are at 854.2 eV and 871.5 eV of Ni^2+^, and 855.8 eV and 872.8 eV of Ni^3+^ for Cu_0.6_Ni_0.4_Co_2_O_4_. Comparing to the peaks for NiCo_2_O_4_, binding energy of Ni 2p for Cu_0.6_Ni_0.4_Co_2_O_4_ are stronger. The peak areas mean the relative content of element valence. It’s noted that the peak area of them is various, which shows the relative content of Ni^2+^ is lower in Cu_0.6_Ni_0.4_Co_2_O_4_. It indicates the amount of Ni^2+^ decreases in the presence of Cu^2+^. There are four peaks in Cu 2p spectrum for CuCo_2_O_4_ and Cu_0.6_Ni_0.4_Co_2_O_4_, indicating copper is divalent. Six peaks of Co 2p for the three materials can be found, manifesting the cobalt is trivalent and divalent. Similar, binding energy of Cu 2p, Co 2p for Cu_0.6_Ni_0.4_Co_2_O_4_ are stronger than CuCo_2_O_4_, and NiCo_2_O_4_, respectively. The peak positions are corresponding to previous reports [32,39,41], indicating that Cu_0.6_Ni_0.4_Co_2_O_4_ is prepared successfully and Cu^2+^ reduces the content of Ni^2+^. The electron density of Cu_0.6_Ni_0.4_Co_2_O_4_ in the outermost shell is affected by doping elements.

Transition metal oxides undergo a conversion reaction in the lithiation, with the change in metal valence and the formation of solid electrolyte interface (SEI) layer and Li_2_O [3]. Figure 6 shows the CV test of NiCo_2_O_4_, CuCo_2_O_4_, and Cu_0.6_Ni_0.4_Co_2_O_4_ anode electrodes for the first three curves at a scan of 0.2 mV s^−1^ between 0.01 and 3 V. For the NiCo_2_O_4_ anode material, during the first scan, the main cathodic peaks locate in 0.7 V corresponding to the reduction of NiCo_2_O_4_ to metallic Ni and Co (Equation (2)), and the broad peaks at ca. 0.37 V corresponds to SEI layer [42]. The anode peaks are at the 1.63 V and 2.3 V, corresponding to oxidation of metallic Ni^0^ to Ni^2+^ and Co to Co^3+^, respectively. (Equations (5), (7) and (8)). For the CuCo_2_O_4_, during the initial scan, the main cathodic peaks locate in 0.75 V, corresponding to the reduction of CuCo_2_O_4_ to metallic Cu and Co (Equation (3)). It is mentioned that there are two peaks at ~0.5 V and ~0.6 V, revealing formation of SEI layer. The anode peaks were at the 1.1 V, 1.5 V and 2.2 V, corresponding to oxidation of metallic Cu and Co to CuCo_2_O_4_ (Equations (6), (7) and (8), respectively). For Cu_0.6_Ni_0.4_Co_2_O_4_, during the first scan, the main cathodic peaks locate in 1.2 V and 0.8 V, assigning to reduction of metal oxide and formation of SEI film (Equation (4)). The anode peaks are at the 1.63 V and 2.3 V, corresponding to oxidation of metallic Ni, Cu and Co to Cu_0.6_Ni_0.4_Co_2_O_4_ (Equations (5)–(8)). At the same time, the peaks of Cu_0.6_Ni_0.4_Co_2_O_4_ are narrower and stronger than those of NiCo_2_O_4_ and CuCo_2_O_4_. This can be due to co-work of Co, Cu, and Ni, which stabilizes the structure and enhance the electronic conductivity in the process of electrochemical reaction, resulting in a longer cycle life and a lower potential polarization. The different positions of the peaks reveal that the REDOX potential can be changed slightly by composition and content of metal oxide in the chemical reaction. The first cathodic scan of the three anode materials, in addition to the reduction of metals, the formation of SEI film and Li_2_O were occurred, so the second anode scan is quite different from the first one [43,44]. While, the second cycle anode scan of the three materials is roughly the same as the first cycle, demonstrating good cycle performance. The total reaction equations during charging and discharging are described as following:(2)$$NiCo_{2}O_{4}+8Li^{+}+8e^{-}\to Ni+2Co+4Li_{2}O$$
(3)$$CuCo_{2}O_{4}+8Li^{+}+8e^{-}\to Cu+2Cu+4Li_{2}O$$
(4)$$Cu_{0.6}Ni_{0.4}Co_{2}O_{4}+8Li^{+}+8e^{-}\to 0.6Cu+0.4Ni+2Co+4Li_{2}O$$
(5)$$Ni+Li_{2}O\to NiO+2Li^{+}+2e^{-}$$
(6)$$Cu+Li_{2}O\to CuO+2Li^{+}+2e^{-}$$
(7)$$Co+Li_{2}O\to CoO+2Li^{+}+2e^{-}$$
(8)$$C\mathrm{oO}+1/3Li_{2}O\to 1/3Co_{3}O_{4}+2/3Li^{+}+2/3e^{-}$$

Figure 7 shows specific capacities of the NiCo_2_O_4_, CuCo_2_O_4_, Cu_0.6_Ni_0.4_Co_2_O_4_ at the current density of 50 mA g^−1^ between 0.01 and 3 V (vs. Li^+^/Li). The first discharge specific capacities of NiCo_2_O_4_, CuCo_2_O_4_, and Cu_0.6_Ni_0.4_Co_2_O_4_ are 950, 901, and 1120 mAh g^−1^, respectively (Figure 7a). The first charge specific capacities of NiCo_2_O_4_, CuCo_2_O_4_, and Cu_0.6_Ni_0.4_Co_2_O_4_ are 785, 633, and 972 mAh g^−1^, there are large specific capacity losses in the first charge/discharge cycle. We attribute them to some irreversible electrochemical reactions, including the SEI film formation, electrolyte decomposition, and so on. This phenomenon can be observed for most anode materials, which leads to the low coulombic efficiency in the first cycle and explains a large difference in the peak position of the reduction peak between the first and the second cycles in the CV test. The NiCo_2_O_4_ electrode shows a specific discharge capacity of 743 mAh g^−1^ in second cycle, and the specific capacity declines to 245 mAh g^−1^ after 50 cycles. The CuCo_2_O_4_ electrode shows a specific discharge capacity of 690 mAh g^−1^ at the second cycle, while the specific capacity is attenuated to 375 mAh g^−1^ after 50 cycles. The Cu_0.6_Ni_0.4_Co_2_O_4_, which can be regarded as a compound of CuCo_2_O_4_ and NiCo_2_O_4_, exhibits the excellent electrochemical performance. The specific discharge capacity of Cu_0.6_Ni_0.4_Co_2_O_4_ is maintained 880 mAh g^−1^ from the second to the fiftieth cycle, and the coulombic efficiency close to 100%. As shown in Figure 7b, the change of the specific capacity of the three samples with the number of cycles is clearly visible. The capacities of NiCo_2_O_4_ and CuCo_2_O_4_ material go through a huge decay after 20 cycles, which is due to material pulverization. Meanwhile, Cu_0.6_Ni_0.4_Co_2_O_4_′s capacity has been well-maintained. The electrochemical stability is significantly improved because of better electronic conductivity of Cu_0.6_Ni_0.4_Co_2_O_4_ and the synergistic effect of the elements. Due to the various insertion voltages for the metal oxides, the other inactive metal oxides can be “volume-buffering reservoir” to release volume stress when the active metal oxide is in a lithiation/delithiation process at a certain voltage [45]. The capacity rate performance of Cu_0.6_Ni_0.4_Co_2_O_4_ was also studied and the results are showed in Figure 7c. The charge/discharge current density gradually increases to 2000 mA g^−1^ and then returns to 50 mA g^−1^. The charge-discharge voltage platform was found to be almost kept same. The stable reversible discharge capacities decrease from 830 to 150 mAh g^−1^ as the current density increases from 200 mA g^−1^ to 2000 mA g^−1^. Furthermore, when the current density is restored to 50 mA g^−1^, the discharge capacity is 780 mAh g^−1^, which is ca. 88% of the initial discharge capacity. These results demonstrate that the Cu_0.6_Ni_0.4_Co_2_O_4_ has a better rate performance and a higher cycling stability.

To further compare the properties of the materials and the modification effects. The charge transfer behaviors of NiCo_2_O_4_, CuCo_2_O_4_, and Cu_0.6_Ni_0.4_Co_2_O_4_ are measured according to electrochemical impedance spectroscopies (EIS). The Nyquist plots of the electrodes, with frequency range between 0.01 Hz to 100 kHz are shown in Figure 8. All plots make up a semicircle and a sloped line, the fitting circuit is shown insert. In the high intermediate frequency region, the diameter of the semicircle indicates charge transfer resistance (*R_ct_*) in the interface between the electrolyte and grains of active material [46]. It is easy to see that the diameter of the semicircle of Cu_0.6_Ni_0.4_Co_2_O_4_ is much smaller than the diameter of NiCo_2_O_4_ and CuCo_2_O_4_. The electron transfer resistance of Cu_0.6_Ni_0.4_Co_2_O_4_ (120 Ω) is less than of NiCo_2_O_4_ (300 Ω) and CuCo_2_O_4_ (380 Ω), which shows Cu_0.6_Ni_0.4_Co_2_O_4_ sample has the lowest charge-transfer resistance. The low-frequency line stands for the Warburg resistance (Wo) related to the lithium ions diffusion in electrode materials. The line slope of Cu_0.6_Ni_0.4_Co_2_O_4_ is larger than of NiCo_2_O_4_ and CuCo_2_O_4_, indicating faster solid-state diffusion of lithium ion is in the Cu_0.6_Ni_0.4_Co_2_O_4_. This evidence further proves that Cu_0.6_Ni_0.4_Co_2_O_4_ has better electrochemical performance and lower charge transfer resistance due to co-work of Cu^2+^, Ni^2+^/Ni^3+^ and Co^2+^/Co^3+^.

## 4. Conclusions

In this work, Cu_0.6_Ni_0.4_Co_2_O_4_ nanowires, with high electrochemical performance, have been synthesized by a simple and scalable hydrothermal approach. The material shows initial specific discharge and charge capacities of 1120 and 972 mAh g^−1^ at current density of 50 mA g^−1^. After 50 cycles, its discharge capacity maintains 880 mAh g^−1^, and the coulombic efficiency is close to 100%. After a high current charge/discharge test, the reversible specific capacity can still be restored to 780 mAh g^−1^ at 50 mA g^−1^, which is ca. 88% of the initial capacity. EIS and CV tests have proved that co-doping with appropriate amount of Cu and Ni could significantly enhance ion diffusion and electron conduction during charge-discharge process. We attribute the high performance to relatively low activation energy for electron transfer between Cu^2+^, Ni^2+^/Ni^3+^ and Co^2+^/Co^3+^, the electronic conductivity and structure stability for Cu_0.6_Ni_0.4_Co_2_O_4_ nanowires are improved greatly. Therefore, it is important to further research into other multi-component transition metal oxides as anode materials in lithium battery.

## Figures and Tables

**Figure 1 nanomaterials-10-00191-f001:**
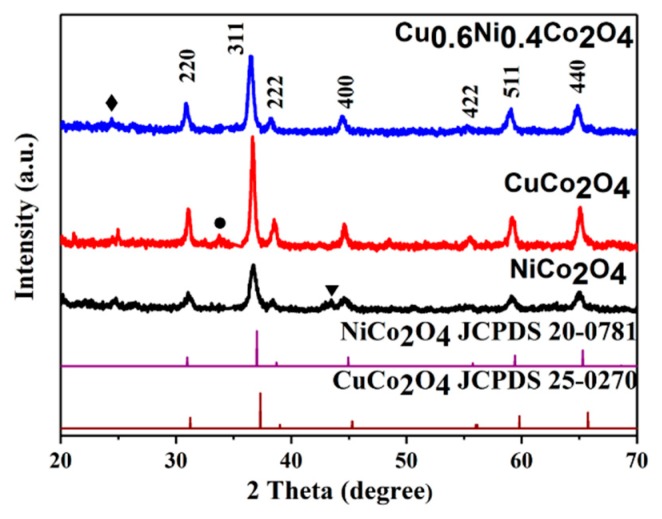
X-ray diffraction (XRD) pattern of the NiCo_2_O_4_, CuCo_2_O_4_, and Cu_0.6_Ni_0.4_Co_2_O_4_ powders.

**Figure 2 nanomaterials-10-00191-f002:**
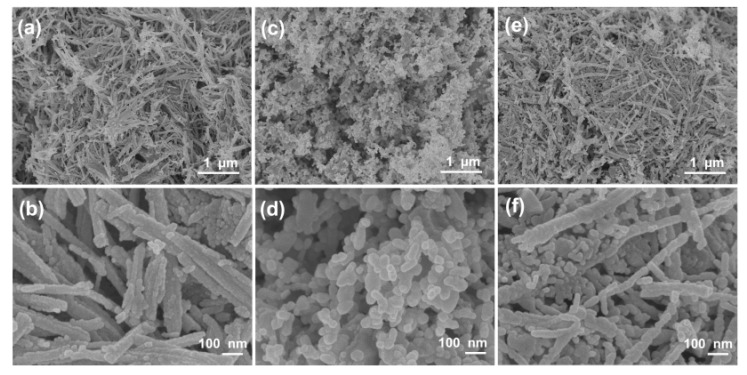
Scanning electron microscopy (SEM) images of (**a**,**b**) NiCo_2_O_4_, (**c**,**d**), CuCo_2_O_4_ and (**e**,**f**) Cu_0.6_Ni_0.4_Co_2_O_4_.

**Figure 3 nanomaterials-10-00191-f003:**
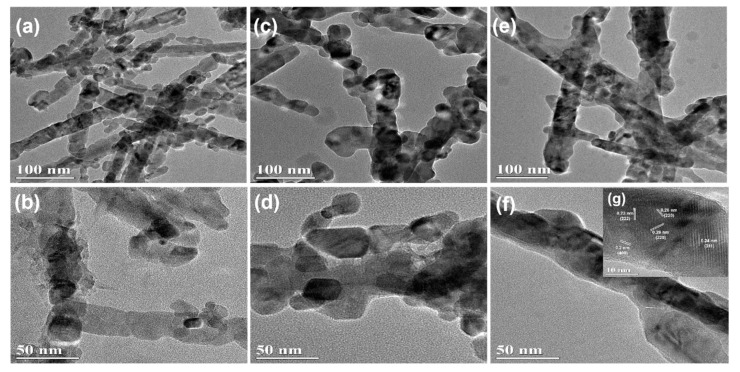
Transmission electron microscopy (TEM) image of (**a**,**b**) NiCo_2_O_4_, (**c**,**d**) CuCo_2_O_4_ and (**e**,**f**) Cu_0.6_Ni_0.4_Co_2_O_4_; (**g**) the inset showing the HRTEM image of Cu_0.6_Ni_0.4_Co_2_O_4_.

**Figure 4 nanomaterials-10-00191-f004:**
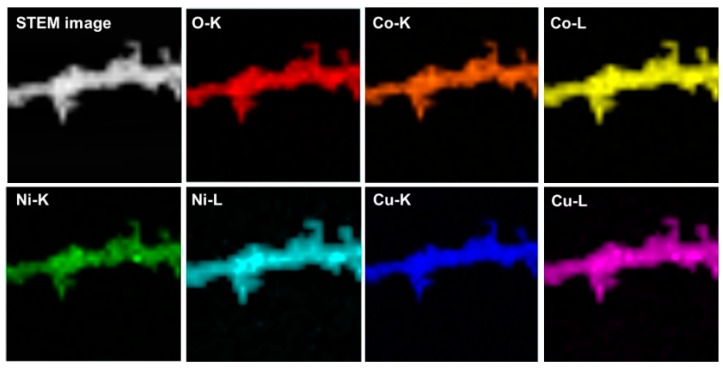
Elementary mapping of Cu_0.6_Ni_0.4_Co_2_O_4_.

**Figure 5 nanomaterials-10-00191-f005:**
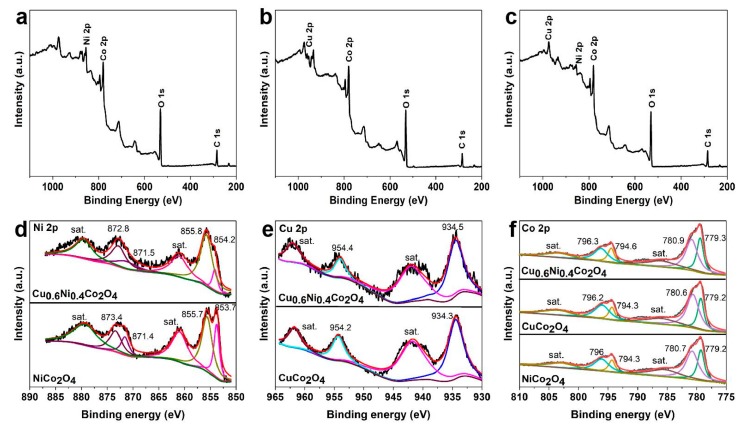
XPS spectra of (**a**) NiCo_2_O_4_, (**b**) CuCo_2_O_4_, (**c**) Cu_0.6_Ni_0.4_Co_2_O_4_ and (**d**) Ni 2p of NiCo_2_O_4_ and Cu_0.6_Ni_0.4_Co_2_O_4_, (**e**) Cu 2p of CuCo_2_O_4_ and Cu_0.6_Ni_0.4_Co_2_O_4_, (**f**) Co 2p of the three materials.

**Figure 6 nanomaterials-10-00191-f006:**
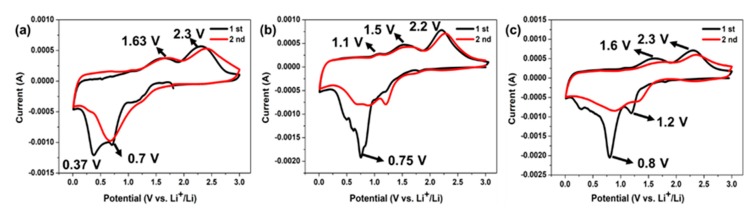
Cyclic voltammetry measure of (**a**) NiCo_2_O_4_, (**b**) CuCo_2_O_4_, and (**c**) Cu_0.6_Ni_0.4_Co_2_O_4_.

**Figure 7 nanomaterials-10-00191-f007:**
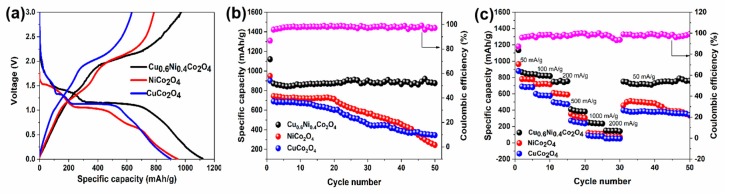
Electrochemical tests of the three samples: (**a**) Initial charge and discharge curves profiles for NiCo_2_O_4_, CuCo_2_O_4_, Cu_0.6_Ni_0.4_Co_2_O_4_ at the current of 50 mA g^−1^ between 0.01 and 3 V (vs. Li^+^/Li). (**b**) Cycling performance of the three powders at a current of 50 mA g^−1^. (**c**) Capacity rate performance of the anode materials.

**Figure 8 nanomaterials-10-00191-f008:**
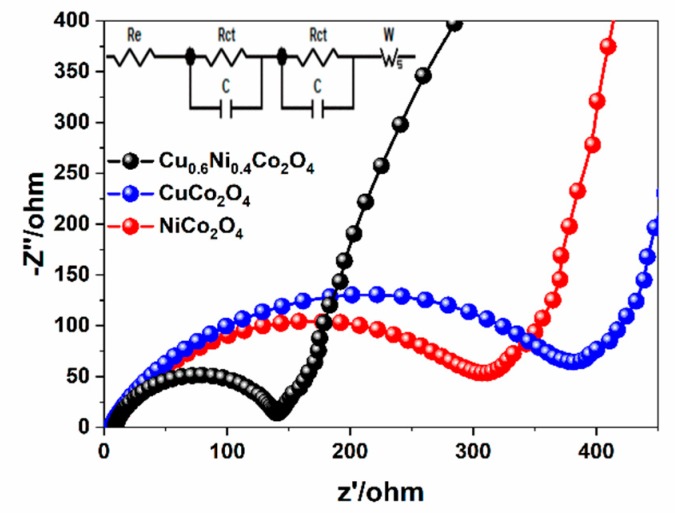
Nyquist plots for the three electrodes in the frequency range from 100 kHz to 0.01 Hz.

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
