# Peer review of "Nonstoichiometric Cu0.6Ni0.4Co2O4 Nanowires as an Anode Material for High Performance Lithium Storage"

_nanomaterials, 2020, doi:10.3390/nano10020191_

Round 1

Reviewer 1 Report

In this manuscript, the authors synthesized nonstoichiometric Cu0.6Ni0.4Co2O4 nanowires and investigated its lithium storage properties. The results indicated that Cu0.6Ni0.4Co2O4 shows better electrochemical performance compared with CuCo2O4 and NiCo2O4 anodes. This is an interesting work and the reviewer believes this manuscript fits the scope of Nanomaterials very well. The reviewer would like to recommend the acceptance for publication after minor revision as described below:

As the author mentioned in the abstract that Cu0.6Ni0.4Co2O4 shows better Li storage performance due to its better electronic conductivity and structural stability. However, the reviewer did not see any experimental results related to such a “structure stability” of Cu0.6Ni0.4Co2O4 anode. What is the main result of XPS spectra as shown in Figure 5? Such as the valance state of Cu, Ni, and Co in the pristine material. Rate performances of CuCo2O4 and NiCo2O4 anodes are suggested to be added in Figure 7c for comparison. As shown in Figure 6, the CV curve of NiCo2O4 shows a peak at about 0.37 V during the initial reduction process. While CuCo2O4 and Cu0.6Ni0.4Co2O4 did not show peaks at such a low voltage. This should be mentioned, and the possible mechanism should be discussed in the revised manuscript. How did the authors obtain the conclusion about the electrochemical mechanism of Cu0.6Ni0.4Co2O4 anode (line 170-171)? Some additional characterizations such as XPS spectra after the initial discharge/charge are suggested to be added.

Author Response

Reviewer 1#:

In this manuscript, the authors synthesized nonstoichiometric Cu0.6Ni0.4Co2O4 nanowires and investigated its lithium storage properties. The results indicated that Cu0.6Ni0.4Co2O4 shows better electrochemical performance compared with CuCo2O4 and NiCo2O4 anodes. This is an interesting work and the reviewer believes this manuscript fits the scope of Nanomaterials very well. The reviewer would like to recommend the acceptance for publication after minor revision as described below:

1. As the author mentioned in the abstract that Cu6Ni0.4Co2O4 shows better Li storage performance due to its better electronic conductivity and structural stability. However, the reviewer did not see any experimental results related to such a “structure stability” of Cu0.6Ni0.4Co2O4 anode.

Response: Thanks for your suggestion. As we known, nanowires are beneficial to the material’s electronic conduction and structural stability, especially to overcome the problem of contact between nanoparticle material and current collectors and/or conductive agents caused by volume expansion during charge and discharge. We did compare the structure of the nanowire and nanoparticle electrodes after cycling, and the SEM images show that the former is significantly more stable.

2. What is the main result of XPS spectra as shown in Figure 5? Such as the valance state of Cu, Ni, and Co in the pristine material.

Response: There are six peaks in Ni 2p spectrum for NiCo2O4 and Cu0.6Ni0.4Co2O4. The peaks are at 854.2 eV and 871.5 eV of Ni2+, and 855.8 eV and 872.8 eV of Ni3+ for Cu0.6Ni0.4Co2O4. Comparing to the peaks for NiCo2O4, binding energy of Ni 2p for Cu0.6Ni0.4Co2O4 are stronger. There are four peaks in Cu 2p spectrum for CuCo2O4 and Cu0.6Ni0.4Co2O4, indicating copper is divalent. Six peaks of Co 2p for three anodic materials can be found, manifesting the cobalt is trivalent and divalent. Similar, binding energy of Cu 2p, Co 2p for Cu0.6Ni0.4Co2O4 are stronger than CuCo2O4 and NiCo2O4, respectively. The result indicates that Cu0.6Ni0.4Co2O4 is prepared successfully and Cu2+ reduces the content of Ni2+.

3. Rate performances of CuCo2O4 and NiCo2O4 anodes are suggested to be added in Figure 7c for comparison.

Response: Thanks for your suggestion. The rate performances of CuCo2O4 and NiCo2O4 are also added in this revised manuscript (Figure 7c)

4. As shown in Figure 6, the CV curve of NiCo2O4 shows a peak at about 0.37 V during the initial reduction process. While the CuCo2O4 and Cu6Ni0.4Co2O4 did not show peaks at such a low voltage. This should be mentioned, and the possible mechanism should be discussed in the revised manuscript.

Response: As we know, Ni has a very high catalytic activity. When the nickel content is too high, it will easy lead to reduction and decomposition of the organic electrolyte at a relatively low potential. As reported in previous literature, the CV curve of NiCo2O4 shows a peak at about 0.37 V during the initial reduction process due to solid electrolyte interface (SEI) film (C. Zhang, J-S. Yu. Chem. Eur. J. 2016, 22, 4422–4430).

5. How did the authors obtain the conclusion about the electrochemical mechanism of Cu6Ni0.4Co2O4 anode (line 170-171)?

Response: Transition metal oxides undergo a conversion reaction in the lithiation, with the change in metal valence and the formation of solid electrolyte interface (SEI) layer and Li2O. The main content of Cu0.6Ni0.4Co2O4 can be seen as CuO, NiO and Co3O4. The electrochemical mechanism of such metal oxide has been reported in previous literature. (P. Poizot, S. Laruelle, S. Grugeon, L. Dupont, J.-M. Tarascon. Nature, 2000, 407, 496-499.)

6. Some additional characterizations such as XPS spectra after the initial discharge/charge are suggested to be added.

Response: Thank you for your suggestion. We will add this part of testing and discussion in our future work.

Reviewer 2 Report

This manuscript describes the development of Cu0.6Ni0.4Co2O4 nanowires as anode materials in Li-ion cells.  These materials have been prepared by a facile hydrothermal method and characterized for their composition and nanostructure using the conventional techniques, XRD, SEM and TEM.   Electrochemical tests show a high capacity of  972 mAh g-1 at current density of 50 mA g-1 for first cycle, and ~98% of which is maintained over 50 cycles, with a coulombic efficiency of 98%. Both the high rate tests and the EIS data indicate that the charge transfer resistance and structural stability are improved with the Cu0.6Ni0.4Co2O4 compared to either CuCo2O4 or NiCo2O4. Even though the specific capacity of this mixed metal oxide nanowire is higher than with graphite, there is poor interfacial stability as evident from a high irreversible capacity (in the first cycle), and also during subsequent cycling (low coulombic efficiency).  This, combined with a high oxidation voltages (>1 V vs Li), make these materials less attractive compared to conventional graphite anode. There is only moderate novelty in this work since the CuCo2O4 or NiCo2O4  have been well studied already and the results are not significantly different with this material. However, based on the interest on high capacity anodes for Li-ion batteries, this study may be sufficiently interesting to the battery technologists and relevant to the journal.  Based on this, I recommend the manuscript be considered for publication after a revision addressing the following comments:

Page 1, line 23: Replace ‘circle’ with ‘cycle’. Page 1, line 34:  “The energy density of LIB is closely related to the lithium storage performance and voltage of the material”.. This applies to both anode and cathode. Page 1 Line 38: .”Different from lithium intercalation mechanism of graphite, Co3O4 based on conversion reaction as anode materials, which makes volume changed greatly and metal reunion”… Not quite clear. Suggest rephrasing Page 1, line 39: . “Co3O4 restricts specific capacity on the verge of theoretical value”.. …this sentence is a bit awkward. Page 2 , line 58: have become widely concerned as anode materials.  doesn’t read well! Page 2 line 69: are significantly improved. Line 90: What is the electrolyte used here? Line 147: “precious researchers” previous researchers? Line 154: Why is there such a large difference in the reduction and oxidation potentials? Are these oxidation and reduction potential are properly assigned?  Looks like there are reduction peaks in the first cycle, from the electrolyte reduction.  It would be appropriate to discuss the redox peaks of these materials in the second cycle. Line 157: the main cathode peaks located in 0.75 V, corresponding to the reduction of CuCo2O4 to metallic Cu and Co. It is mentioned that there are two peaks at ~0.5 V and ~0.6 V, revealing CuCo2O4 unstable in a chemical reaction.” Not clear! Line 159: “1 V, 1.5 V and 2.2 V, corresponding to oxidation of metallic Cu and Co to CuCo2O4 (eq. (2))”….. Why three peaks? Line 166: “The first anodic scan of the three anodic materials, in addition to the reduction of metals, the…” Shouldn’t this be cathodic? 167: And the second cathodic….” Shouldn’t be this be anodic? Looks like the cathodic peak around 0.8 V is related to the electrolyte reduction (and SEI formation), correct? What is the electrolyte here? Line 190: The irreversible capacities are considerably high (~300 ,Ah/g).  With this high irreversible capacity, it is impossible to get any improvement in the specific energy for the cell (with cathodes being the Li source). The coulombic efficiency is also quite low (98%) here. Does it imply a poo interfacial stability? With this low coulombic efficiency, it is difficult to get a meaningful cycle life.

Author Response

Thanks for your suggestion, please see the reply attachment

Reviewer 3 Report

Comments to the authors

The paper entitled " Nonstoichiometric Cu0.6Ni0.4Co2O4 nanowires as an anode material for lithium storage " submitted for publication in Nanomaterials gathers characterization and electrochemical performance of nonstoichiometric Cu0.6Ni0.4Co2O4 nanowires used as an anode material for lithium storage. The authors found a specific capacity of 880 mAh g-1 and a coulombic efficiency about 98% after 50 cycles. Though these values are promising and many techniques of characterization have been carried out, the manuscript presents a succession of results description without scientific comments. Moreover, some erroneous points deserve to be corrected.

Without being exhaustive, different aspects that must be improved are listed below:

XRD characterization: “It is discovered that the characteristic peaks of the NiCo2O4 and CuCo2O4 are very similar”: that is not a discovery as CuCo2O4 and NiCo2O4 structures have already been established (JCPDS 25-0270 and 20-0781). “This is rational because Cu6Ni0.4Co2O4 is a composite 106 of NiCo2O4 and CuCo2O436“: this comment does not provide any information. It would have been preferable to determine and compare the cell parameters of the three compounds.” No characteristic peaks related to the impurity“: there are different small peaks that are not indexed while the authors claim that there is no impurity. They must clarify this point. There is confusion between Figures 3 and 4 in the text. The distribution of SEM images is not advisable for comparison. it would have been better to make a line for the 1μm scale and a second below for the highest magnification. Figure 2d highlights larger nanowires in the case of CuCo2O4. Comments are waiting for on this subject. TEM images (Fig3a to Fig3e) do not provide additional information with respect to SEM images. Moreover fig3f must be improved. “Additionally, the molar ratio of Cu to Ni to Co well matches the designed value”: the element contents must be added in the text. XPS measurements are not complete and the corresponding comments are very unconvincing “was different from those of NiCo2O4 and CuCo2O4”. Comparison of XPS spectra of Cu 2p, Ni 2p and Co 2p between the three compounds should be presented and discussed. The part of electrochemical properties investigated by cyclic voltammetry must be improved and enriched by further structural characterization, at least during the first cycle, to give piece of evidence of the proposed mechanism. Whatever the cycle number, there are at least to peaks during discharge and charge processes. Which kind of species can be found in the electrodes at different points of the cycle? As is, the mechanism cannot be validated. Besides, references about SEI formation are different (40, 41 on the one hand, 9, 42 on the other hand). The authors must check that. The authors write “The electrochemical stability is significantly improved because of better electronic 192 conductivity and structural stability of Cu6Ni0.4Co2O4.”: what is the scientific proof of this assessment? EIS explanations must be clarified: radius of diameter of semi-circle means Rct? Interface between electrolyte and grains of active material must be discussed instead of Interface between electrolyte and electrode; frequencies must be added on the figure; the Nyquist diagram shape at low frequencies is different for Cu6Ni0.4Co2O4 nanowires electrode: what are the explanations?

Author Response

Thansks for your suggestion, please see the reply attachment.

Round 2

Reviewer 3 Report

Comments to the authors

Among the various points justifying the wish that the paper be rejected, some were discussed in the authors' response without impact in the text while others resulted in more or less satisfactory modifications in the text.

Each of them are discussed below, considering the authors’answers.

XRD characterization: “It is discovered that the characteristic peaks of the NiCo2O4 and CuCo2O4 are very similar”: that is not a discovery as CuCo2O4 and NiCo2O4 structures have already been established (JCPDS 25-0270 and 20-0781). “This is rational because Cu6Ni0.4Co2O4 is a composite 106 of NiCo2O4 and CuCo2O436“: this comment does not provide any information. It would have been preferable to determine and compare the cell parameters of the three compounds.” No characteristic peaks related to the impurity“: there are different small peaks that are not indexed while the authors claim that there is no impurity. They must clarify this point.

Response: Indeed, only from XRD data can’t completely confirm that we synthesized pure Cu0.6Ni0.4Co2O4, so we further demonstrated this conclusion through XPS and element mapping. Although there are some very tiny peak from the XRD data, we attribute them to a larger noise signal.

This answer does not give any element of response and is not convincing at all. Moreover, no modification has been made in the text. The authors cannot assess that peaks obtained around 25, 33 or 44° are attributed to a larger noise signal!

There is confusion between Figures 3 and 4 in the text.

Response: Thanks, we have corrected them this time.

Ok with this change

The distribution of SEM images is not advisable for comparison. it would have been better to make a line for the 1μm scale and a second below for the highest magnification. Figure 2d highlights larger nanowires in the case of CuCo2O4. Comments are waiting for on this subject.

Response: We have revised it as your suggestion in this version manuscript. Thanks very much.

Ok, SEM images have been moved for easier comparison BUT no mention and no discussion about the diameter size of nanowires have been added as required. Besides, SEM images provide no information on microstructures as written in the first line of the paragraph. When authors write “all these nanowire are assembled ..”, does it mean “connected”? Syntax should also be improved.

TEM images (Fig3a to Fig3e) do not provide additional information with respect to SEM images. Moreover, fig3f must be improved.

Response: The TEM images (Figures 3a-d) of three materials have further proved their microstructures, such as NiCo2O4 and Cu0.6Ni0.4Co2O4 are nanowire assembled from primary particle, and CuCo2O4 is pure nanoparticle, and these prove that these materials are solid rather than hollow structure, the Figure 3f has been improved in this manuscript.

Authors should differentiate between microstructure and morphology. SEM images show the morphology of the three samples without any doubt. TEM images do not “further proved that the microstructures are nanowires”. In Figure 3f, points still don’t match with the red plotted circles. The quality of the SAED pattern needs to be improved.

“Additionally, the molar ratio of Cu to Ni to Co well matches the designed value”: the element contents must be added in the text. XPS measurements are not complete and the corresponding comments are very unconvincing “was different from those of NiCo2O4 and CuCo2O4”. Comparison of XPS spectra of Cu 2p, Ni 2p and Co 2p between the three compounds should be presented and discussed.

Response: Thanks for your suggestions. We have tested the proportions of Cu, Ni and Co in the material using the ICP method, and the results (Cu:Ni:Co=0.41:0.61:1.99) show that the molar ratio of Cu/Ni/Co well match the designed value. At the same time, we have compared the XPS spectra of Cu 2p, Ni 2p and Co 2p in Figures 5a-c, and discussed in page 7 (red font part).

Content analysis by ICP is a good way to verify the stoichiometry of the compound. Besides, there is probably an inversion between Cu and Ni in the results reported above, right? It could have been a good idea to report them in the manuscript.

Figures 5d, 5e and 5f have been modified. It is clearer now but the size has to be increased, it is difficult to observe the different contributions. Moreover, the same color should be used for peaks corresponding to the same energy (for example at 878 eV for Ni 2p).

In the discussion, authors write that peaks are “different from those of NiCo2O4 and CuCo2O4”. Explanations are required. The whole added paragraph should be checked: for Ni 2p and Co 2p, six peaks are mentioned while only four of them are listed both in text and in figure 5d; more details must be given when authors write “the peak area of them is various”, what does it mean?; “Cu2+ reduces the content of Ni2+”: ICP analysis are probably more significant than the very low shift in energy of XPS peaks.

The part of electrochemical properties investigated by cyclic voltammetry must be improved and enriched by further structural characterization, at least during the first cycle, to give piece of evidence of the proposed mechanism. Whatever the cycle number, there are at least to peaks during discharge and charge processes. Which kind of species can be found in the electrodes at different points of the cycle? As is, the mechanism cannot be validated.

Response: For transition metal oxides as anode materials in lithium-ion batteries, there is a high irreversible capacity because of side effects, such as formation of SEI film, electrolyte decomposition, and so on. The peaks at ~1.6 V and ~ 2.35 V can be assigned to Ni0 to Ni2+, and Co0 to Co2+ and Co3+. (S. Abouali, Akbari Garakani M , Z. L. Xu, J.-K. Kim. Carbon, 2016, 102:262-272.) The peak at ~1.1 V can be assigned to Cu0 to Cu2+, that why there are three peaks in the CV curve of CuCo2O4. (S. Sun, Z. Wen, J. Jin, Y. Cui, Y. Lu. Microporous and Mesoporous Materials, 2013,169, 242-247.). There are seven equations related to the conversion reaction between this material and metallic lithium are shown in the list (page 8, line 196-197).

The part of electrochemical properties investigated by cyclic voltammetry must be improved and enriched by further structural characterization, at least during the first cycle, to give piece of evidence of the proposed mechanism: this point has not been treated. Such characterizations are necessary to validate the proposed mechanism, all the more so as the galvanostatic cycles shape tends to indicate firstly insertion reaction followed by conversion process. The authors’ answer is not sufficient.

The mechanism during the second and further cycles is not discussed at all to support the proposed mechanism. Corresponding cycles are not presented in figure 7a… This electrochemical is not satisfactory.

Besides, references about SEI formation are different (40, 41 on the one hand, 9, 42 on the other hand). The authors must check that.

Response: We have removed the reference 9 and 42, Thanks.

OK

The authors write “The electrochemical stability is significantly improved because of better electronic 192 conductivity and structural stability of Cu6Ni0.4Co2O4.”: what is the scientific proof of this assessment?

Response: We attribute the high performance of Cu0.6Ni0.4Co2O4 to its better electronic conductivity and structural stability, the reasons as following:

Appropriate amount of doped Cu2+, which can be reduced to copper during discharge process, which can increase the electronic conductivity of the material. Comparing with Cu6Ni0.4Co2O4 nanoparticle, nanowire structure of Cu0.6Ni0.4Co2O4 has an advantage of electron conduction. Nanowires have large open space, which can buffer volume expansion during charge/discharge process, so the electrode structure fabricated by the material will be more stable in structure. As we known, nickel will accelerate the decomposition of the electrolyte, and partial replacement of Ni by Cu will beneficial to reduce side reactions between the electrode material and electrolyte.

EIS explanations must be clarified: radius of diameter of semi-circle means Rct? Interface between electrolyte and grains of active material must be discussed instead of Interface between electrolyte and electrode; frequencies must be added on the figure; the Nyquist diagram shape at low frequencies is different for Cu6Ni0.4Co2O4 nanowires electrode: what are the explanations?

Response: According to the literature, the diameter of semicircle has a positive correlation with charge transfer resistance (Rct). (S. Sun, X. Zhao, M. Yang, L. Wu, Z. Wen, X. Shen. Scientific Report, 2016, 6, 19564). The low-frequency line stands for the Warburg resistance (Wo) related to the lithium ion diffusion in electrode materials. The line slope of Cu0.6Ni0.4Co2O4 is larger than of NiCo2O4 and CuCo2O4, indicating faster solid-state diffusion of lithium ion is in the Cu0.6Ni0.4Co2O4 (J. Yang, et al., Nano Research, 2016, 9, 612). A detailed discussion on this part is already provided in this revised manuscript.

I agree with authors that have changed radius by diameter of semicircle, but I still disagree with the discussion about the Warburg region. Obviously, at low frequencies, there is not only one slope that can be observed. It's probably not as simple as the authors write. The frequencies of the points dispersed on the line with a 45° slope should be indicated, compared. The explanation of the Nyquist diagrams still must be improved.

Author Response

Thank you for your suggetions. I have carefully revised the manuscript again this time,and hope this time can meet your requirments. see attachment (light blue font)
